# A Decade of Treatment of Canine Parvovirus in an Animal Shelter: A Retrospective Study

**DOI:** 10.3390/ani10060939

**Published:** 2020-05-29

**Authors:** Kevin Horecka, Steve Porter, E. Susan Amirian, Ellen Jefferson

**Affiliations:** Research Department, Austin Pets Alive!, Austin, TX 78703, USA; steve.porter@austinpetsalive.org (S.P.); ea25@rice.edu (E.S.A.); ellen.jefferson@austinpetsalive.org (E.J.)

**Keywords:** canine parvovirus, animal sheltering, treatment, veterinary epidemiology, survival analysis

## Abstract

**Simple Summary:**

The canine parvovirus (CPV) is a highly contagious gastrointestinal disease which affects unvaccinated, insufficiently vaccinated, or improperly vaccinated dogs and results in a fatality rate greater than 90% if left untreated. Treatment in private practice settings can often cost several thousand dollars, making it an unaffordable option for many pet owners as well as a challenging population to treat for shelters. Here, we examine 11.5 years of data from Austin Pets Alive!, a private animal shelter in Austin, TX, which has treated 5127 dogs infected with CPV since 2008. We show an 86.6% (*n* = 4438/5127) survival rate, with the most critical period of treatment during the first five days of care, and detail the protocols used to achieve this high proportion of successful treatment outcomes. A CPV season was observed peaking in May and June and accounting for as much as a 41 animal/month increase compared to low periods in August, September, December, and January. Low-weight animals and male animals were found to be at higher risk for mortality. Together, these results aim to assist shelters in creating programs to treat this disease and to inspire future research into improving practices in treatment and prevention.

**Abstract:**

Here, we present 11.5 years of monthly treatment statistics showing an overall intake of 5127 infected dogs between June 2008 and December 2019, as well as more detailed datasets from more recent, less protracted time periods for the examination of mortality risk, seasonality, and resource requirements in the mass treatment of canine parvovirus (CPV) in a private animal shelter. The total survival rate of animals during the study period was 86.6% (*n* = 4438/5127 dogs survived) with the probability of survival increasing to 96.7% after five days of treatment (with 80% of fatalities occurring in that period). A distinct parvovirus season peaking in May and June and troughing in August, September, December, and January was observed, which could have contributed as much as 41 animals peak-to-trough in the monthly population (with a potential, smaller season occurring in October). Low-weight and male animals were at higher risk for death, whereas age was not a significant contributing factor. Treatment time averaged 9.03 h of total care during a seven-day median treatment duration. These findings, taken together, demonstrate that canine parvovirus can be successfully treated in a sustainable manner within a shelter setting using a largely volunteer workforce.

## 1. Introduction

Canine parvovirus (CPV) is a highly contagious and relatively common virus that causes substantial morbidity and mortality in dogs worldwide [1,2]. Symptoms include acute enteritis, fever, and potential cardiac sequelae (i.e., myocarditis and myocardial fibrosis), with the most clinically severe disease manifesting (i.e., sepsis and severe dehydration) among dogs infected between six weeks and six months of age [2,3]. Survival rates are dependent upon treatment, with estimates ranging from 9% in untreated populations up to 80%–90% in tertiary care centers [2,4,5]. In private practice settings, the cost of treatment can be $1000 to $2000 (USD), indicating that financial constraints may be a factor in disease-related euthanasia [6,7].

Rather than providing treatment, nonprofit rescues and shelters may often opt to euthanize infected dogs, both due to financial constraints and as a potential disease control tactic, given the highly contagious nature of the illness. Austin Pets Alive! (APA!), a nonprofit, private shelter in Austin, Texas, espouses the philosophy of saving the animals most at risk for euthanasia for treatable/manageable conditions (a so-called, No Kill philosophy) [8,9,10,11]. Since 2008, APA! has implemented a protocol to treat CPV-infected dogs in a quarantine environment [12], averaging several hundred dogs a year. CPV-infected dogs are treated within a section of the shelter exclusively designated for the treatment of parvovirus with the support of volunteers and donations, which, along with optimized mass treatment protocols and procedures, aids in reducing costs to a reported average of between $56 and $300 USD (depending on the precise method of calculation; i.e., the inclusion of volunteer time and days on site in cost estimates) [13,14] and approximately 9 h of volunteer time per animal.

Treatment for CPV in this study varied somewhat based on symptom presence and severity, animal weight (with 5 lbs or 2.27 kg used as a threshold as decided by Veterinary staff), and the result of the most recent CPV test. Although almost all animals in the intensive care unit (ICU) test positive at intake, additional protocols are present for critically exposed animals (i.e., members of a litter whose siblings, with which they are cohoused, show symptoms) that would have been euthanized at their originating shelter but have yet to present symptoms or otherwise tested negative. These negative tests may contradict prior evidence from the originating shelter (i.e., a prior positive test). An exposed treatment (ET; described in Appendix A) was implemented until these animals presented with symptoms (or, until they pass a five-day hold and show two consecutive solid stools along with an additional negative test, which is required of all animals leaving the ICU). More symptomatically severe or at-risk animals were given intravenous (IV) treatments (a more time-consuming treatment protocol which is more difficult to implement with healthy animals as they are more likely to remove their IV catheters), while less symptomatically severe and lower-risk animals were given subcutaneous (SQ) treatments. These treatments were subject to change given changing status of the animal during its stay. Additionally, it is important to note that all animals had unknown vaccination statuses and were generally assumed to be unvaccinated, improperly vaccinated, or to have undergone incomplete vaccination courses.

To our knowledge, Austin Pets Alive! has treated more dogs with CPV than any other organization listed in the bibliography, and, as such, is among the largest samples of this population in the world. Given this experience in CPV treatment, an examination of its protocols, success rates, signalments, seasonality trends, and resource usages is warranted, providing vital data that can inform the development of guidelines for in-shelter CPV treatment and management. In the present study, we review the 11.5 years of history of this innovative treatment program using four data sets, with increasing temporal and feature resolution, which have been gathered as the data collection practices of the shelter have improved. Taken together, this work strongly supports the assertion that canine parvovirus is an imminently treatable condition, even in persistently resource-restricted environments.

## 2. Materials and Methods

Data in this study are sourced from four data sets (see Table 1 for a summary) as the data collection methods of the shelter improved over time. Initially (2008–2013), APA! collected data only in month-by-month aggregates, as is common for newer shelters with limited resources, and this collection continued through 2019 such that Monthly Aggregate data (N = 5127) was available from 2008–2019. Additionally, in 2013, an End-of-Shift Report (N days = 2474) was created that then allowed volunteers and, eventually, staff to enumerate the animals treated, their condition, the hours per treatment shift, and the number of volunteers and staff that participated in treatment. These data are still being collected at the time of publication. In 2016, data on specific animals, including their signalment information such as weight, sex, approximate age, and specific intake dates, became available; these data are also still being collected as of the time of publication. Finally, treatment-level data (i.e., collected during twice a day treatment; see the treatment protocols in Appendix A for more information) is present for animals from a specific time period of January 2017 through January 2018.

The following statistical analyses (Table 2, right column) were used to examine key questions (Table 2, left column) in regard to the data sets (Table 2, middle column. The Python [15] language was used with critical libraries including SciPy [16,17], NumPy [18,19], Pandas [20], Scikit-Posthocs [21], Lifelines [22], Matplotlib/Seaborn [23,24], and FBProphet [25].

Wherever possible, descriptive statistics (sample size, mean, and/or median) were examined. Kolmogorov–Smirnov test for goodness of fit as well as Anderson–Darling test were used to examine normal and Left–Gumbel distribution fits in the Monthly Aggregate (N = 127; 2008–2019) and Treatment Records (N = 589; 2017–2018) data while a Kaplan–Meier curve was used to examine survival over time in the Treatment Records data. Although it might normally be preferable to perform a one-way ANOVA for detection of seasonal variation, assumptions of normality of residuals (as tested via a Shapiro–Wilk test) and homogeneity of variance (as tested via Brown–Forsythe) revealed that this test would be inappropriate. As a result, a Kruskal–Wallis H-test, a nonparametric test for the population medians across months being equal, was employed. Additionally, although day-level resolution is present in the End-of-Shift Reports (N days = 2474; 2013–2020), these daily values will include duplicate animals and are, therefore, nonindependent. To correct for this, the twice a month median values of these days for each month were taken before performing analyses at the month level. This is somewhat conservative given the average length of stay is approximately seven days. This reduced the test variance, but substantially improved the independence of samples. Although more complex models could be used to assess the relationship of signalment variables (sex, weight, and age) to outcome, in this case, direct tests of these key signalment variables based on prior hypotheses (justified by prior work) are employed in an attempt to provide the most clarity into their relationship with survival.

A few noteworthy caveats exist within these data sets that deserve enumeration before examining the results of the analyses. Firstly, in the Monthly Aggregate data, data are missing for the months of April 2016 through December 2016 (though a year-level survival rate exists which matches the existing trends and can be used to assess the overall survival rate). This data loss was due to a change in shelter management platform software and is, therefore, not suspected to create unintended biases in the results. Secondly, the End-of-Shift Reports contain daily counts of numbers of animals present in the ICU, but these animals are not identified and, as such, special care must be taken in analyzing these data as day-to-day measurements are not independent. Thirdly, two major changes in treatment protocol occurred throughout this period: (1) the addition of Force Feeding as a part of the standard protocol occurred in December of 2016, and (2) the addition of high-dose IV Vitamin C was initiated in February 2018.

## 3. Results

As per the questions and analyses in Table 2, the results follow:

### 3.1. Survivability

#### 3.1.1. Overall Survival Rates

In examining the survival rates, we primarily examined the distribution of survival rates on a monthly level as this data set spans the entire 11.5 years of operational history from 2008 to 2019. There were four notable dips in survival such that the rate was less than 70% in a given month. Three of those were survival rates of 66.6%, while one was 0% as only a single animal was present during that month and it did not survive treatment. Excluding the 0% outlier, the Kolmogorov–Smirnov test for goodness of fit of the data to a normal distribution was significant (rejecting the null hypothesis that the data is sampled from this distribution, i.e., the data is non-normal and “greater” than the normal distribution; KS = 0.159, *p* = 0.001). As such, a Left–Gumbel distribution (selected via an Anderson–Darling test) [31] was fit to assess (KS = 0.066, *p* = 0.317, i.e., we fail to reject the null hypothesis and assume the data can be fit to this distribution) the descriptive statistics of the survival rates. These descriptions can be found in Figure 1. Note that although this analysis can be useful in understanding expected variability in this outcome measure, the critical statistic was not found via taking the means of the survival rates, but rather via taking the overall number of survived animals divided by the overall population (4438/5127 = 86.6%).

#### 3.1.2. Time Course of Survival

We can examine the time course of survival (Figure 2) using the Treatment Records (N = 589; 2017–2018), primarily to illustrate how critical the first five days of treatment are for animals with CPV infections. If animals survive the first five days, the probability of survival increases from 85.6% on intake to the shelter to 96.7% after the 10th treatment (end of day five). Note that the average overall length of stay is 14.33 treatments or just over seven days. The peak death rate occurs on the 7th treatment and 80% of deaths are accounted for by the first 10 treatments (i.e., first five days).

#### 3.1.3. Symptomaticity

The Treatment Records data (N = 589; 2017–2018) is the only data set of the ones examined in this study with sufficient feature richness to be used to assess symptom severity. Within the 589 animals in the Treatment Records data, 574/589 (97.5%) became symptomatic with CPV (where symptomaticity is defined as having one of: inappetence, vomiting, ill formed stool, diarrhea or bloody diarrhea, pale gum color, and lethargy or coma). Of the 15 asymptomatic patients, two were discharged early (after three and four treatments, respectively). Six were distemper-exposed animals that were held for longer durations. The remaining six were held for the typical duration due to extensive exposure to CPV positive animals. Of the 574 animals that were symptomatic of CPV, 230/574 (40%) developed severe symptoms (where severe is defined as the presence of bloody diarrhea, six or more incidents of vomiting in a 12 h period, white or gray gums, and/or lethargy/coma). Of the 230 dogs with severe symptoms, 70 (30.4%) did not survive. For comparison, of the 344 dogs with milder symptoms, 15 (4.4%) did not survive.

#### 3.1.4. Population Versus Survival Rates

There was no evidence the overall Population is related to the Survival Rate (*p* = 0.20; F = 1.692). This indicates that as population increases/decreases, no corresponding increase/decrease was seen in survival.

### 3.2. Seasonal Trends

In order to assess seasonality within the data, we first examined if the population medians of all groups were equal across months in the Monthly Aggregate (N = 5127; 2008–2019) and End-of-Shift Report (N days = 2474; 2013–2020) data sets (see Figure 3 for a visualization of distributions). Both data sets violated assumptions of normality of residuals (W = 0.95, *p* < 0.001 for Monthly Aggregate and W = 0.90, *p* < 0.001 for End-of-Shift Report via Shapiro–Wilks) but not homogeneity of variance (F = 1.06, *p* = 0.40 for Monthly Aggregate and F = 0.46, *p* = 0.93 for End-of-Shift Report via Brown–Forsythe). A Kruskal–Wallis H-test, a nonparametric test for the population medians being equal, was performed. Monthly Aggregates were significantly different from the median (KW = 19.72, *p* = 0.049) and End-of-Shift Report months were significantly different from the median (KW = 29.32, *p* = 0.002) suggesting there are month-over-month population differences.

Follow-up Dunn’s test with False Discovery Rate correction revealed that May and June were the peak months when compared to January, December, August, and September (*p* < 0.05 for all after FDR correction; see Figure 4 for details). All of this points to the idea that there was at least one peak CPV season in the summer (May–June) with a trough immediately after (September). These tests, however, were quite conservative.

To get more resolution on the effect, a follow-up Fourier Coefficients model was produced using the Monthly Aggregate data to examine the relative effect component sizes of a CPV season, as well as the nonstationarity of the population as the organization grew. Two models were trained with one and two coefficients (i.e., seasonal peaks), and the cross-validated mean absolute error (MAE) was compared for a 180 day period with up to a one-year horizon, 42 days was chosen as the benchmark horizon. The Mean Absolute Error was lower for the model with two coefficients (23.04 vs. 25.03 for a single-component model), suggesting it is a marginally better fit (i.e., a smaller, second CPV season may be present; see Figure 4). Both models produced yearly trend components with increases in population year-over-year accounting for four additional animals per month. The primary CPV season had a peak-to-peak interval of 41 animals. The interaction of these components was not directly examined.

### 3.3. Signalment Variable Relationships

Three key signalment variables were hypothesized to influence outcomes based on prior work: age, weight, and sex. Age and weight were examined directly in the Animal Records (N = 1957; 2016–2020) data via a Welch’s T-test and a significant effect of weight (t = −4.287, *p* < 0.001, d = 0.396) was found, but no significant effect of age (t = 0.815, *p* = 0.415) was found. Note that weight was undefined for 444/1957 animals, and age was undefined for 7/1957. Sex was examined using a Fisher’s exact test, and a significant effect of sex was found, with males dying more frequently at (N = 210/931) 22.6% of the time and females dying (N = 114/996) 14.0% of the time (see Table 3 for contingency table for this sample). Sex was undefined for 27/1957 animals.

### 3.4. Capacity for Care

Finally, in order to aid other organizations in implementing programs like the one discussed in this paper, we examined the amount of staff and volunteer time needed to treat animals in the ICU. We used the total number of animals divided by the shift time multiplied by the number of people on shift in the End-of-Shift Report (N days = 2474; 2013–2020) data to find a unimodal distribution with mean of 0.63 h per animal and median of 0.5 h per animal (See Figure A1 and Figure A2 in Appendix B for distributions relevant to these analyses). When the mean time per treatment was multiplied by the mean number of treatments required for animals in the Treatment Data, we found animals required, on average, 9.03 h of treatment across the duration of their stay. As both of these distributions are skewed, the median treatment time was lower than the mean value.

## 4. Discussion

Although previous work has established effective treatment practices [32,33,34] and discussed the potential influence of signalment factors on survival [35,36,37,38], the implementation of CPV treatment programs has yet to become common practice for animal shelters (though some exceptions exist) [39,40]. Here, we present 11.5 years of CPV treatment with increasing feature resolution as well as the protocols for operation of a CPV ICU in an animal shelter setting (see details in Appendix A and Appendix C). We demonstrate an 86.6% survival rate (4438/5127 = 86.6%), which is higher than many previous reports [4,41,42,43]. The most critical period for survival was the first five days, during which 80% of fatalities occurred, suggesting rapid care is needed in order to effectively treat CPV. A distinct seasonal trend, peaking in the summer in May and June and troughing in August, September, January, and December, suggesting the presence of a possible second, smaller CPV season in addition to the dominant, summer season which represented an approximately 41 animal peak-to-trough difference in population. Finally, low-weight animals and male animals were at higher risk of death than other populations.

It is worth noting that the variability in survival rate was surprisingly limited across the datasets evaluated here. In particular, when populations increase, one might expect a reduction in survival rate due to insufficient staffing or other resources, which reduce quality of care, but no evidence for such an effect is present in these data. Moreover, with a single anomalous exception, no single month had a survival rate below 66.6% despite the presence of large disaster events such as Hurricane Harvey [44] and the Bastrop Fires [45]. This survival rate was almost certainly biased upward when compared to an untreated survival rate of 9% [2,3], as many animals only developed mild symptoms (potentially due to maternal antibodies [46] or partial vaccination protection [47]); however, given that standard procedure at many shelters is to kill exposed animals, the effective survival rate of the population would be 0% in many shelters, regardless of their chances of surviving CPV itself. As the program has expanded (at a rate of approximately four additional animals per month), taking in animals from wider regions of Texas, the efficiency of a dedicated treatment ICU for CPV has become apparent. Future work should attempt to examine if different regions have similar incidence of the illness, seasonality, and signalment correlates of survival, as some of these effects could be specific to the Central Texas region.

Consistent with other work [36,48,49,50,51], the presence of one or two CPV seasons (with the larger peak occurring around May–June) can drastically impact the need for staffing and resources. Note that a second CPV season, though weakly supported by these data, was consistent with volunteer and staff reports. Despite our data showing no evidence for population-dependent changes in survival rate, a large (41-animal, peak to trough) swing in population strains staffing and can increase turnover. As a result, maintaining a large, well-trained volunteer organization was critical to stable operations (see Appendix C for more detail on this). The reason for the presence of a CPV season is, as of yet, unknown. Some work has examined wider relationships between disease spread in animal populations relative to factors such as temperature, rainfall, and resource availability [52,53], but no such examination has been conducted specifically for CPV. Attempts to perform this examination with the data in this study would require additional information about the sources of animals, which was not present in the data set and is beyond the scope of this study. Future work should examine potential natural and public-health drivers which might contribute to this seasonality. As CPV is so effective at surviving in the environment for extended durations [54], one might speculate that persistent reservoirs of the virus exist within the environment [55,56,57], to which unvaccinated, insufficiently vaccinated, or improperly vaccinated animals are being seasonally exposed (i.e., parks, rivers, or even animal shelters). Note that the authors have not seen a single case example of an animal in the custody of Austin Pets Alive! in the general shelter population breaking with CPV in the entire history of the organization, likely due to aggressive vaccination and quarantine procedures [58,59].

Three major factors were examined in this study that were hypothesized to relate to the survival probability include the time since intake, weight, and sex of the animal. These factors have been examined in previous work [36,37,58]. Though each illustrates important risk factors for CPV survival, time since intake, in particular, highlights the criticality of additional interventions being developed that can be implemented in the first few days after arrival at the shelter (with survival of the first five days increasing survival rate by 11.5%). Many owners may wait until symptoms have progressed to the extreme (i.e., bloody diarrhea, lethargy, and severe vomiting), making the treatment less likely to be effective. Although some other work has also assessed the contribution of breed to survival [59], we were unable to assess relationships to breed in this study as the breeds of animals were entirely subjectively evaluated, visually, by volunteers and staff. Future work may examine if breed or developmental differences may impact survival in a shelter setting. Other work has not been able to link weight/size to survival in CPV [36,37], potentially due to the limited size and scope of their data. This study cannot fully assess the importance of weight, however, as the treatment protocols differ according to weight groups (see Appendix A for details). That being said, both in the anecdotal accounts of the veterinarians, staff, and volunteers as well as in the Treatment Data, we see a significant relationship between weight and survival that cannot be disregarded (in particular as the more extensive treatment is given to the population which survives less; i.e., low-weight animals). Future work should attempt to more carefully address this risk factor. Sex, on the other hand, showed similar male-biased mortality patterns (with an 8.6% difference) as has been previously observed both in CPV [37] as well as across huge varieties of species and ailments [60,61,62]. One final factor which was not examined in this study due to difficulty in diagnosis was the presence of comorbidities that might impact survival. Canine distemper, for example, often appears alongside CPV and can drastically reduce the chances of survival [2,63], though no direct study of this relationship is known by the authors. Special care is taken in the ICU to create air-breaks via plastic sheeting and variable risk isolation rooms to attempt to prevent these sorts of comorbidities, but outbreaks and singleton events can, and do, occur (though their frequency is difficult to assess given these data as only the exposure status, not the distemper symptoms, were tracked and no distemper tests were administered). Future work should more closely examine methods of reducing the spread of, especially, upper respiratory infections.

Although we believe the inferences discussed in this study are on firm statistical ground, some weaknesses in the data, including the lack of treatment-level resolution across the entire time range, gaps in some portions of the data set, and difficulty in measuring every variable that otherwise might be of interest, are worth consideration. Weight (in the Animal Records data; N = 1957; 2016–2020) was a particularly impoverished measure as weighing animals on intake has not always been standard practice. Continued data collection (especially given the significant improvements in shelter software systems in recent years) could help clarify some details in this study, such as if there is a second CPV season, if there are other trends in the survival of high-risk populations, and if there are efficiency gains that could be exploited to optimize treatments.

Finally, a few interesting future directions exist in treatment that should be examined in a shelter setting. Continuous 24 h monitoring via automated, low cost, noncontact medical devices [64], fecal transplants [65], antiviral treatments [66,67], blood transfusions [68], and more community outreach to encourage early intervention could potentially help save the last 10%–15% of animals in this population. Regardless of these innovations, together, these results demonstrate the imminently practical treatability of canine parvovirus in an animal shelter setting.

## 5. Conclusions

For most of the history of the canine parvovirus at the time of this publication, this disease has been a death sentence for infected dogs, whether due to its expense and difficulty in treatment, or due to the active euthanasia of animals with the infection by organizations fearing spread to the entire population. Although these fears are in some ways warranted, as CPV is incredibly contagious and deadly when left unchecked and untreated, by following the practices vetted over the last 10 years by Austin Pets Alive! (see Appendix A for details), shelters should expect to successfully save >80% of CPV-infected animals in their care with little risk to their general populations. The total survival rate of animals during the 11.5-year period of this study was 86.6% (n = 4438/5127 dogs survived) with the probability of survival increasing to 96.7% after five days of treatment. A distinct parvo season peaking in May and June and troughing in September and January was observed, which contributed as much as 41 animals peak-to-trough in the monthly population. Low-weight and male animals were at higher risk for death, whereas age was not a significant contributing factor. Further research should examine these high-risk populations to determine how to best provide them care, but canine parvovirus should now be viewed as imminently and practically treatable within the animal shelter setting.

## Figures and Tables

**Figure 1 animals-10-00939-f001:**
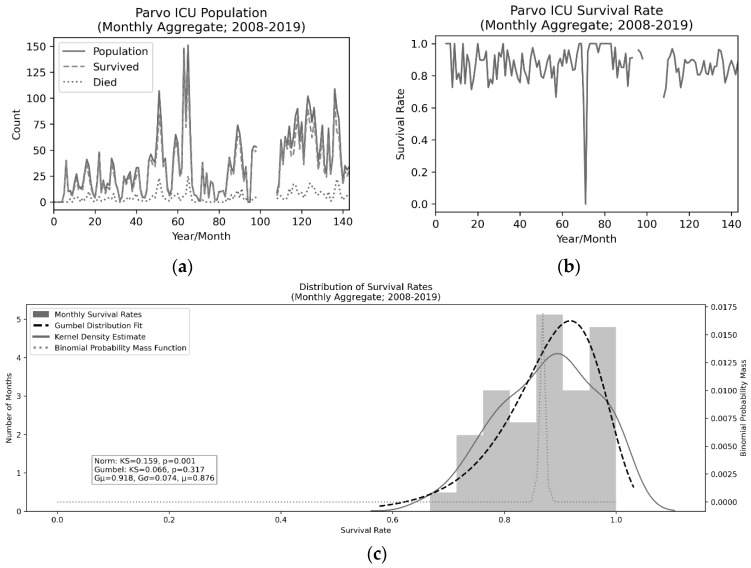
Population values (**a**), survival rates (**b**), and distributional properties (**c**) of canine parvovirus (CPV)-infected dogs from July 2008 to December 2019. The expected survival rates are approximately 86.6% and there is no evidence for a relationship to the overall population occupying the intesive care unit (ICU).

**Figure 2 animals-10-00939-f002:**
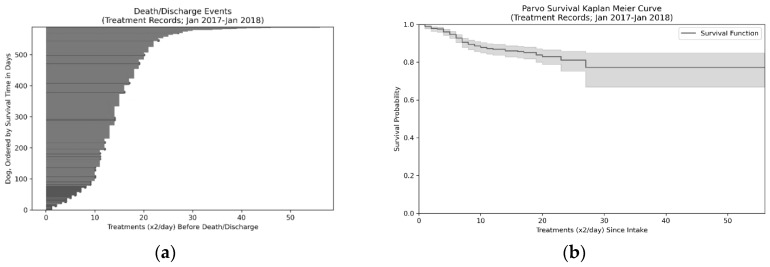
Event plot (**a**) and Kaplan–Meier curve (**b**) for survival over time (**c**) for Canine parvovirus (CPV)-infected dogs. Darker lines in the death/discharge events plot (a) represent animals who did not survive. Distribution of Survived and Died outcomes by treatment shift can be seen in panel (c) with 80% of deaths occurring on or before the 10th treatment, which, at two treatments/day, is on the 5th day.

**Figure 3 animals-10-00939-f003:**
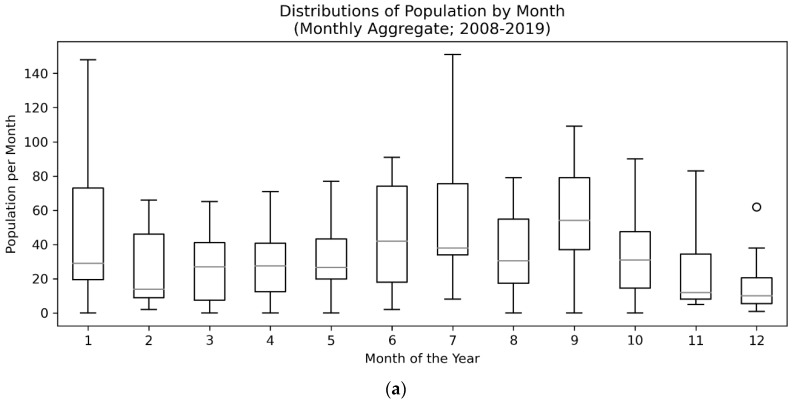
Box and whisker plots of Monthly Aggregate (N = 5127; 2008–2019) (**a**) and End-of-Shift Report (N days = 2474; 2013–2020) (**b**) data showing clear seasonal trends in the number of Canine parvovirus (CPV)-infected dogs peaking in the summer months. Circles indicate outlier points with interquartile range greater than 1.5 (i.e. Q3 + 1.5 × (Q3–Q1) for the upper bound).

**Figure 4 animals-10-00939-f004:**
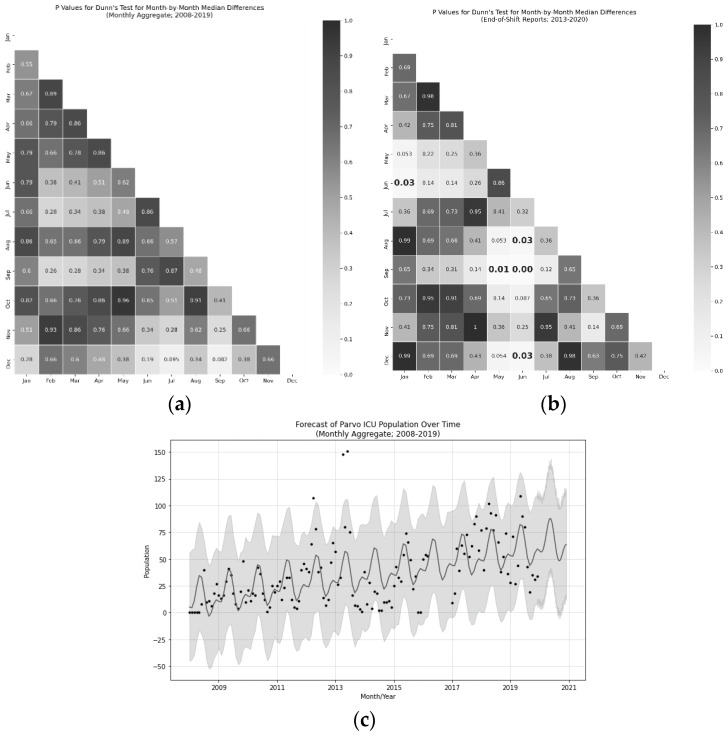
Heatmaps (**a**,**b**) of Dunn’s test p-values (FDR corrected) for Monthly Aggregate (N = 5127; 2008–2019) (**a**) pairwise differences and End-of-Shift Report (N days = 2474; 2013–2020) (**b**) pairwise differences from the monthly median canine parvovirus (CPV)-infected population. Bolded elements indicate statistical significance with alpha less than 0.05. Forecast visualization (**c**) from Fourier components model via FBProphet. Peaks can be observed in seasonality in May and June and troughs can be seen in January, August, and September. The overall population of CPV-infected dogs in the ICU increased at a rate of approximately four animals per month with a seasonal component accounting for approximately 41 dogs between peak and trough season.

**Table 1 animals-10-00939-t001:** The data sets, time ranges, and temporal resolutions for the four data sets examined in this study.

Data Set	Date Range	Resolution	Dogs (N)
A. Monthly Aggregate	2008–2019	Survived/Died Populations	5127
B. End-of-Shift Reports	2013–2020	Daily Population/Staffing	N/A (2474 days)
C. Animal Records	2016–2020	Individual Animals	1957
D. Treatment Records	2017–2018	Twice-Daily Treatments Per Animal	589

**Table 2 animals-10-00939-t002:** Key questions, their associated data sets, and statistical methods employed. An explanation of what each test is used for is provided for convenience.

Question	Data Set(s) Used	Statistical Analysis	Statistical Analysis Description
1. What is the expected distribution of survival rates when treating canine parvovirus (CPV)?	A	Descriptive Statistics, Kolmogorov–Smirnov test [26], Gumbel distribution fit [27],	KS test is used for goodness-of-fit of different distributions (like Gumbel)
2. How likely is survival after each day of treatment?	D	Kaplan–Meier Curve Estimation [28,29]	KM Curves describe survival over time
3. Is there evidence for a relationship between the number of CPV-infected dogs being treated and their survival rate?	A	Linear Regression	Linear regression examines simple linear relationships between variables
4. Is there evidence for seasonal trends in the number of CPV-infected dogs in the shelter each month?	A, B	Kruskal–Wallis H-test [30] and Dunn’s test, Fourier Coefficient Forecast Model (FBProphet) [25]	KW and Dunn’s examine the hypothesis of monthly differences, FBProphet quantifies seasonal effects
5. Is there evidence for signalment variables influencing the survival rate for subpopulations of CPV-infected dogs (i.e., age, weight, and sex)?	C, D	Welch’s T-Test, and Fisher’s Exact Test	T-Test and Fisher’s Exact directly compare outcomes for variables of interest
6. How many days should CPV-infected dogs be expected to require treatment? How many hours of care does this translate to for staff or volunteers?	B, D	Descriptive Statistics, Linear Regression	Linear regression examines simple linear relationships between variables

**Table 3 animals-10-00939-t003:** Contingency table for effect of sex on survival, with male dogs showing 22.6% more fatalities than female dogs as measured in the Animal Records data (N = 1957; 2016–2020).

	Male	Female
Survived	721	882
Died	210	114

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
