# Peer review of "A Decade of Treatment of Canine Parvovirus in an Animal Shelter: A Retrospective Study"

_animals, 2020, doi:10.3390/ani10060939_

Round 1

Reviewer 1 Report

Comments

The paper entitled “A decade of treatment of canine parvovirus in an animal shelter: a retrospective” is a very interesting research, very innovative and it contains interesting data. However, I would like to make suggestions to increase the understanding of the study.

The survival rate is very good. Considering that perhaps this result is not reached even by private structures, I suggest to describe your structure and organization of the structure very well. The authors indicate to visit the website but the paper should be autonomous.

Raw and easier data, especially descriptive, are missing, while there are more advanced and less immediate analyses. The reader must first make his own idea with simple, more immediate descriptive data and then he can read secondary analyses. Probably the authors should report the main descriptive statistics for each databases.

Specific  comment

Line 9 and along the paper:  “Parvo” as short form to indicate “parvovirus”, “parvovirus  infected “ or similar is too confidential for a scientific journal.

Line 10: …affects especially unvaccinated dogs but not only; often the disease appears in not completely vaccinated dogs or not correctly vaccinated (and for an owner they are “vaccinated”).

Generally references are not put in the abstract.

Line 63-64: it's probably true, but the authors can't be so absolute. A professor of mine said: - Never say that you are the first to do something because there is certainly someone in the world who did it before you!

You can write “the largest among the known bibliography”  or “among the largest…”

Line 82: By order of appearance this appendix should be A and the other B.

First line of Table 1: range 2008-2019 or 2008-2013?

Line 88: what is “visa vi”?

Line 99: explain the acronym ICU and describe in details the structure and its organization. Probably this aspect is the key point for obtaining your good results. It is not obvious to have this type of structure in a shelter.

Line 106: the first results to be reported are the descriptive data, the frequencies of survivors and deaths, with indications of the range of years or similar (for example the percentage at the line 120 should be described initially; the same for the table 3). Then, the chapter 3.3 Signalment variable relationship could be appropriate and after it, the others.

There are two chapter 3.3.

Is there a graph / analysis showing how survival has improved over the years? Probably by virtue of an implementation of the protocol over time. Maybe the graph of the Forecast analysis, but it should be better explained.

The blocks of figures are generally divided into sections in a scientific paper: a), b) and c) and in general they should be explained better in the captions. Figures and tables should be autonomous, so the phrase "This figure corresponds to Table ..., Question ...! “ is not appropriate.

Figure 2, block three: Number of animals on the right should be on the left?

Table B1: it is not clear. If animals are asymptomatic, how do I think they have parvovirus? Do you do an incoming screening? By what criteria? To any dog entering? It should be specified. I don’t understand the differences in the list.

Author Response

The paper entitled “A decade of treatment of canine parvovirus in an animal shelter: a retrospective” is a very interesting research, very innovative and it contains interesting data. However, I would like to make suggestions to increase the understanding of the study.

The survival rate is very good. Considering that perhaps this result is not reached even by private structures, I suggest to describe your structure and organization of the structure very well. The authors indicate to visit the website but the paper should be autonomous.

> Some relevant details behind the structure of the organization have been added to an Appendix C and a reference to this section has been provided inline in the text.

Raw and easier data, especially descriptive, are missing, while there are more advanced and less immediate analyses. The reader must first make his own idea with simple, more immediate descriptive data and then he can read secondary analyses. Probably the authors should report the main descriptive statistics for each databases.

> Additional descriptive statistics for each data set have been added every time a data set is initially mentioned to provide better clarity including the N and date range. Additionally, more extensive descriptive statistics of symptomaticity was added to a section 3.1.3 which should allow for a more concise understanding of some aspects of the population. The specific values in Table 3 as well as in the main survival rate value have been added.

Specific  comment

Line 9 and along the paper:  “Parvo” as short form to indicate “parvovirus”, “parvovirus  infected “ or similar is too confidential for a scientific journal.

> Changed to “parvovirus”, “CPV”, or “CPV infected” throughout

Line 10: …affects especially unvaccinated dogs but not only; often the disease appears in not completely vaccinated dogs or not correctly vaccinated (and for an owner they are “vaccinated”).

> Changed to “affects unvaccinated, insufficiently vaccinated, or improperly vaccinated” in both the Simple Summary and Abstract, and added callout of this issue in the discussion (paragraph 2, second 2 last sentence)

Generally references are not put in the abstract.

> Removed the references in the abstract

Line 63-64: it's probably true, but the authors can't be so absolute. A professor of mine said: - Never say that you are the first to do something because there is certainly someone in the world who did it before you!

You can write “the largest among the known bibliography”  or “among the largest…”

> Softened the language in the last paragraph. It now reads: “Austin Pets Alive! has treated more dogs with CPV than any other organization listed in the bibliography, and, as such, is among the largest samples of this population in the world. Given this experience in CPV treatment…”

Line 82: By order of appearance this appendix should be A and the other B.

> Appendix B and A were swapped

First line of Table 1: range 2008-2019 or 2008-2013?

> I believe this confusion was caused by the beginning of the first Materials and Methods paragraph in which the monthly aggregate was called out as being 2008-2013. The monthly aggregate was collected then, and continued collection after as other metrics were added. The phrasing here was confusing, so it was changed to: “Initially (2008-2013), APA! collected data only in month-by-month aggregates, as is common for newer shelters with limited resources, and this collection has continued through today such that monthly aggregate data is available from 2008-2019. Additionally, in 2013, an End-of-Shift Report was created “

This hopefully clarifies the idea that data collection was enhanced in 2013, again in 2016, and again in 2017.

Line 88: what is “visa vi”?

> This was changed to “in regard to” for better clarity

Line 99: explain the acronym ICU and describe in details the structure and its organization. Probably this aspect is the key point for obtaining your good results. It is not obvious to have this type of structure in a shelter.

> added “intensive care unit” for clarity before the first use of the acronym and added details on the ICU operation in appendix C

Line 106: the first results to be reported are the descriptive data, the frequencies of survivors and deaths, with indications of the range of years or similar (for example the percentage at the line 120 should be described initially; the same for the table 3). Then, the chapter 3.3 Signalment variable relationship could be appropriate and after it, the others.

There are two chapter 3.3.

> Capacity for Care section was changed to 3.4

Is there a graph / analysis showing how survival has improved over the years? Probably by virtue of an implementation of the protocol over time. Maybe the graph of the Forecast analysis, but it should be better explained.

> Figure 1 contains the survival rates over time. There is no relationship between time and survival or population and survival. The population finding is listed in section 3.1.3.

The blocks of figures are generally divided into sections in a scientific paper: a), b) and c) and in general they should be explained better in the captions. Figures and tables should be autonomous, so the phrase "This figure corresponds to Table ..., Question ...! “ is not appropriate.

> Letter labels for subfigures were added. Additional descriptions were added to each figure caption to make them more autonomous. 

Figure 2, block three: Number of animals on the right should be on the left?

> This was corrected.

Table B1: it is not clear. If animals are asymptomatic, how do I think they have parvovirus? Do you do an incoming screening? By what criteria? To any dog entering? It should be specified. I don’t understand the differences in the list.

> More detail regarding the negative test animals has been added to the appendix. A negative test can occur under two main circumstances: first, the animal could be a member of a litter or part of an exposed population and was therefore considered “parvo exposed” such that the shelter from which the animal is being sourced would euthanize the animal and the animal must be isolated from the general population in the ICU (this is a relatively rare occurrence). Second, an animal could have tested positive at a second shelter, but upon intake to APA!, they tested negative (i.e. one of the tests was wrong; also a rare occurrence). We do not have data on all of these test-related elements, but the protocol exists in case these situations arise. 

Reviewer 2 Report

Well written paper presenting important trends in the survival rate of canine parvo across time and the seasonal outbreak. There are some questions that should be addressed or justified before publication. These mainly stem from more information about the study sample should be provided. Also, it would be better to discuss the applicability of treating canine parvo in a shelter based on results.

  • Line 30: ‘is 86.6% (i.e. 4438 dogs survived)…’. Better to present like ‘is 86.6% (n= 4438/5127 dogs survived)’
  • Introduction: It would be better to add a short paragraph describing the recommended treatment plan for canine parvo. This can be linked to your 3 treatment protocols in Appendix B and the logic of your treatment selection.
  • Line 88: ‘visa vi’. Do you mean  ‘Vis-à-vis’ (face to face)?
  • Line 88: ‘(Table 2, middle column.’    ‘ ) ’ is missing.
  • Table2: Regarding the statistical method for question 5, isn’t multivariate regression analysis more suitable? Please justify.
  • Results: What was the distribution of the population of Appendix B, Table B1? It might be possible that the data was biased toward groups with negative Parvo test results or groups with milder symptoms, leading to the high survival rate.
  • Line 240-243: Were there more puppies being transferred into the facility during those two parvo seasons? 
  • Discussion: Would you please provide some arguments based on your results and previous literature to discuss the applicability (medical, financial, and practical in terms of the workforce) of early treatment of canine parvo in a shelter? Better to include this information in the conclusion too.
  • Appendix B, Table B1: If the dogs were tested negative without typical signs of canine parvo, how did you define the dogs were exposed to the disease and should undergo the exposed treatment protocol?
  • Line 372-373: The link of reference 4 is broken. Please provide a new link.

Author Response

Well written paper presenting important trends in the survival rate of canine parvo across time and the seasonal outbreak. There are some questions that should be addressed or justified before publication. These mainly stem from more information about the study sample should be provided. Also, it would be better to discuss the applicability of treating canine parvo in a shelter based on results.

Line 30: ‘is 86.6% (i.e. 4438 dogs survived)…’. Better to present like ‘is 86.6% (n= 4438/5127 dogs survived)’

> This has been corrected

Introduction: It would be better to add a short paragraph describing the recommended treatment plan for canine parvo. This can be linked to your 3 treatment protocols in Appendix B and the logic of your treatment selection.

> This paragraph has been added to the introduction. A special note was also made per another reviewer’s suggestion around understanding the nature of “negative tests” in the context of an intake. Although the situation is incredibly rare, the protocol does support the pre-emptive treatment of exposed animals that have yet to present with symptoms. 

Line 88: ‘visa vi’. Do you mean  ‘Vis-à-vis’ (face to face)?

> changed this to “in regard to” for better clarity and correctness

Line 88: ‘(Table 2, middle column.’    ‘ ) ’ is missing.

> Apologies, I’m unable to find the location in reference. Could you provide some surrounding context so I can locate this error?

Table2: Regarding the statistical method for question 5, isn’t multivariate regression analysis more suitable? Please justify.

> Although we considered more complex models for analyzing the impact of various signalment variables on survival such as multivariate methods or Cox proportional hazard models, we believed in this case that straight-forward tests of the 3 hypotheses in question would provide the most clarity for potential clinical readers. We mention in the discussion that prior work has shown sex biases in survival for CPV and given the protocols around weight in our study, this also felt like an appropriately direct hypothesis to test. Age does significantly relate to weight, but the relationship is fairly weak in our data (and did not seem interesting enough to report). Interaction effects could be potentially interesting in future work that examines these signalment variables (as well as other measures of overall condition severity) in more detail. In particular, fecal status at intake, vomiting at intake, and appetite are all variables we are interested in exploring in future work.

To help clarify this choice for readers, we added the following to section 3.3 on signalment variables: “Although more complex models could be used to address these hypotheses, in this case, direct tests of these key signalment variables based on prior hypotheses are intended to provide the most clarity into their relationship with survival.”

Results: What was the distribution of the population of Appendix B, Table B1? It might be possible that the data was biased toward groups with negative Parvo test results or groups with milder symptoms, leading to the high survival rate.

> Added some descriptive statistics around the number of animals that showed mild, moderate or severe symptoms and definitions therein in a new section 3.1.3 “Symptomaticity” to address these potential concerns. Unfortunately, this issue is complicated by the fact that the shelter has no control over when an animal is brought to them for treatment. Many times, a single animal in a litter is gravely ill while other members are just breaking with symptoms or have not yet broken with symptoms (or even, in the extreme case, tested negative despite significant exposure or a prior positive test). The poor latency with which the public and other shelters generally bring CPV infected animals to seek care is observed by the staff and volunteers of the ICU, but quantifying this effect is quite challenging (and we do not believe our current dataset can fully address this issue). Moreover, it is unclear if maternal antibodies benefit some animals resulting in less severe symptoms. Future studies could/should specifically address these questions.

Line 240-243: Were there more puppies being transferred into the facility during those two parvo seasons? 

> In general, APA!’s population comes from other shelters. Parvo animals can occasionally come to the shelter via a fast-tracked system developed with the local municipal shelter to avoid spreading the disease by forcing them to first go to the municipal shelter before coming to APA!. APA! is not an open-intake shelter, however, so every animal is being sourced from another shelter on some level (almost always by taking them off of the euthanasia lists of those shelters). So the short answer is “absolutely”, it is an increase in parvo at other shelters driving the effect. The degree to which the animals are coming from the greater Austin area versus other more rural shelters that APA! sources animals from is still unclear (and a somewhat difficult question to answer in general given the current protocols and practices in place).

Discussion: Would you please provide some arguments based on your results and previous literature to discuss the applicability (medical, financial, and practical in terms of the workforce) of early treatment of canine parvo in a shelter? Better to include this information in the conclusion too.

> I’m afraid I don’t understand what is being asked here, could you clarify a bit please? Is this specifically in reference to “early treatment” in regard to the finding about the first 5 days being critical? Or is more information about shelter operations what is being sought (for instance, putting the number of Parvo animals into the wider context of the intakes at APA!)? Happy to provide more information if you can help me understand!

Appendix B, Table B1: If the dogs were tested negative without typical signs of canine parvo, how did you define the dogs were exposed to the disease and should undergo the exposed treatment protocol?

> These animals were exposed by being in a litter or co-housed with animals that were both symptomatic and tested positive for Parvo for an extended period. Some additional clarity on this was added in the introduction.

Line 372-373: The link of reference 4 is broken. Please provide a new link.

> This link was fixed.

Reviewer 3 Report

Title is incomplete. Needs ‘study’ or ‘evaluation’ at the end

Flips between tenses: make sure past tense is used throughout eg abstract, line 145

Abstract: would be good to see a little less background and a little more of the findings: what were the successful treatments. It is unusual to include references in an abstract

Introduction: it is worth mentioning that this is in unvaccinated populations. In the UK, parvo is routinely vaccinated against so is not commonly seen unless people opt not to vaccinate puppies

Line 57: remove ‘see results’

Line 105: remove ‘As per the questions and analyses in Table 2, the results follow:’

Line 141: needs test statistic

Line 144: please move justification of stats tests and tests used to the methods section

Line 166-170: mentions significant results but no P values

Line 221-222: ‘one might expect a reduction in survival rate’ please provide a supporting reference

Given the comprehensive results section and level of analysis, it would be preferable to se more discussion based around these findings

Conclusion needs to link more closely to the key findings. The 80% survival isn’t directly discussed in the paper. It reads more as a lay summary than a conclusion

Author Response

Title is incomplete. Needs ‘study’ or ‘evaluation’ at the end

> Added the word “study” to the end of the title

Flips between tenses: make sure past tense is used throughout eg abstract, line 145

> Reviewed all tenses to ensure past tense was used; please let us know if we missed anything

Abstract: would be good to see a little less background and a little more of the findings: what were the successful treatments. It is unusual to include references in an abstract

> Removed the references from the abstract and reworded some sentences to focus more on the findings and less on the background

Introduction: it is worth mentioning that this is in unvaccinated populations. In the UK, parvo is routinely vaccinated against so is not commonly seen unless people opt not to vaccinate puppies

> Some clarification on this was added in a new paragraph in the introduction which details the treatment procedures more. For clarity, APA! does not generally know the vaccination status of animals it intakes, but it is assumed they are generally unvaccinated. Unfortunately, in Central Texas, it is all too common for individuals to ignore vaccination procedures despite veterinary recommendations (or to never see a veterinarian at all).

Line 57: remove ‘see results’

> Removed

Line 105: remove ‘As per the questions and analyses in Table 2, the results follow:’

> Removed this and similar statements throughout

Line 141: needs test statistic

> Corrected this and fixed a rounding error caught in the process

Line 144: please move justification of stats tests and tests used to the methods section

> A new paragraph was added to the methods which succinctly justifies the methods. Corresponding justifications within the results were removed.

Line 166-170: mentions significant results but no P values

> Clarified the meaning of the tests here; all had p<0.05 after FDR correction

Line 221-222: ‘one might expect a reduction in survival rate’ please provide a supporting reference

> To our knowledge, there is no evidence for this in the literature and it has not been studied. It is a common criticism by shelter practitioners that increased population leads to reduced quality of care and more deaths. Some clarifying language was added for this, now reading: “In particular, when populations increase, one might expect a reduction in survival rate due to insufficient staffing or other resources which reduce quality of care, but no evidence for such an effect is present in these data.”

Given the comprehensive results section and level of analysis, it would be preferable to se more discussion based around these findings

> Some additional discussion was added including modification of the end of the first paragraph highlighting the key finding: “...CPV can be treated practically and sustainably in a shelter setting via the protocols laid out in Appendix A and Appendix C. In fact, the survival rate of animals in this 11.5 year history (4438/5127 = 86.6%) was higher than many previous reports.” with associated citations. 

Conclusion needs to link more closely to the key findings. The 80% survival isn’t directly discussed in the paper. It reads more as a lay summary than a conclusion

> The conclusion was revised to include all of the most critical findings and some of the more lay summarization was removed. The critical additions include: “...The total survival rate of animals during the 11.5-year period of this study was 86.6% (n=4438/5127 dogs survived) with the probability of survival increasing to 96.7% after 5 days of treatment. A distinct parvo season peaking in May and June and troughing in September and January was observed which contributed as much as 41 animals peak-to-trough in the monthly population. Low weight and male animals were at higher risk for death while age was not a significant contributing factor. Further research should examine these high-risk populations to determine how to best provide them care, but canine parvovirus should now be viewed as imminently and practically treatable within the animal shelter setting.”

Round 2

Reviewer 1 Report

The paper entitled “A decade of treatment of canine parvovirus in an animal shelter: a retrospective” has been significantly improved. I would like thank the authors for the clarifications.

Some minor mistakes remained.

In my opinion a comment on the possibility to have false negative results by rapid screening test could be appropriate.

The explanation of the existence of false negative result by rapid test could be introduced or near the lines 65-66 or in the appendix A, section A.3 (near lines 404-412): rapid tests for CPV are generally specific but not enough sensitive and a rate of false negative results in infected dogs can arrive to 50% (i.e. Desario et al., 2005; Proksch et al., 2015; Faz et al., 2017). This could be an useful information for the reader, especially if not confident with the test.

The first appearance of the acronym ICU is currently at the line 62.

Line 70, 73, and so on: capital letter for Intravenous or Subcutaneous is not necessary. The same at the line 404 for Negative.

Generally the dosages of the treatments are internationally calculated on kg and not on lb. A change in the text and the tables would make reading easier.

Author Response

The paper entitled “A decade of treatment of canine parvovirus in an animal shelter: a retrospective” has been significantly improved. I would like thank the authors for the clarifications.

Some minor mistakes remained. 

> An additional review of the manuscript for minor issues was conducted. Several minor issues were corrected, but please let us know if you find any specific issues we can correct and we will be very happy to do so.

In my opinion a comment on the possibility to have false negative results by rapid screening test could be appropriate.

The explanation of the existence of false negative result by rapid test could be introduced or near the lines 65-66 or in the appendix A, section A.3 (near lines 404-412): rapid tests for CPV are generally specific but not enough sensitive and a rate of false negative results in infected dogs can arrive to 50% (i.e. Desario et al., 2005; Proksch et al., 2015; Faz et al., 2017). This could be an useful information for the reader, especially if not confident with the test.

> This is a great callout and excellent references to add. In particular, the 2015 study is extremely relevant. Appendix A.3 was changed as suggested and the latter sentences now read:

“Unfortunately, no extensive data exists for the relative success of the CPV test in use at APA!, but the IDEXX ELISA SNAP Parvo Test (the test in use for the entirety of the data covered by this study) claims to have 100% specificity and sensitivity (with 94% and 89% lower bounds on the 95% confidence levels) [69]; however more recent studies have indicated that this could be drastically altered by strain and circumstance, resulting in a high incidence of false negatives (as poor as 50%) [70, 71, 72]. Practitioners should carefully weigh examination of clinical condition and exposure status against negative test results when determining infection status.”

The first appearance of the acronym ICU is currently at the line 62.

> Fixed

Line 70, 73, and so on: capital letter for Intravenous or Subcutaneous is not necessary. The same at the line 404 for Negative.

> Fixed

Generally the dosages of the treatments are internationally calculated on kg and not on lb. A change in the text and the tables would make reading easier.

> Done